RESEARCH COMMUNICATION

# A generalized theory of age-dependent carcinogenesis

**Andrii Rozhok[1]\*, James DeGregori[1,2,3,4]\***

[1]Department of Biochemistry and Molecular Genetics, University of Colorado Anschutz Medical Campus, Aurora, United States; [2]Department of Immunology and Microbiology, University of Colorado Anschutz Medical Campus, Aurora, United States; [3]Department of Pediatrics, University of Colorado Anschutz Medical Campus, Aurora, United States; [4]Department of Medicine, Section of Hematology, University of Colorado Anschutz Medical Campus, Aurora, United States

**Abstract** The Multi-Stage Model of Carcinogenesis (MMC), developed in the 1950 s-70s, postulated carcinogenesis as a Darwinian somatic selection process. The cellular organization of tissues was then poorly understood, with almost nothing known about cancer drivers and stem cells. The MMC paradigm was later confirmed, and cancer incidence was explained as a function of mutation occurrence. However, the MMC has never been tested for its ability to account for the discrepancies in the number of driver mutations and the organization of the stem cell compartments underlying different cancers that still demonstrate nearly universal age-dependent incidence patterns. Here we demonstrate by Monte Carlo modeling the impact of key somatic evolutionary parameters on the MMC performance, revealing that two additional major mechanisms, aging-dependent somatic selection and life history-dependent evolution of species-specific tumor suppressor mechanisms, need to be incorporated into the MMC to make it capable of generalizing cancer incidence across tissues and species.

**Editorial note:** This article has been through an editorial process in which the authors decide how to respond to the issues raised during peer review. The Reviewing Editor's assessment is that all the issues have been addressed (see decision letter).

DOI: https://doi.org/10.7554/eLife.39950.001

**\*For correspondence:**
andrii.rozhok@ucdenver.edu (AR);
james.degregori@ucdenver.edu (JDG)

**Competing interests:** The authors declare that no competing interests exist.

## Introduction

The theoretical foundations of the modern MMC originate from early observations by Fisher and Hollomon (*Fisher and Hollomon, 1951*) and Nordling (*Nordling, 1953*) that the age distribution of cancer death rates follows the sixth power of age. Nordling (*Nordling, 1953*) then formulated the theory and postulated that cancer develops as a sequence of mutations that transform normal cells into malignant cells. Later, Armitage and Doll (*Armitage and Doll, 1954*) supported this idea and proposed a mathematical model to substantiate it. As nothing was known about stem cells and the organization of tissue renewal at the time, Armitage and Doll's model explained the probability of sequential mutation accumulation from a single cell perspective, being unaware of the effects of clonal selection on these probabilities. Armitage and Doll's model, as they acknowledged, also required that the rate of mutation accumulation is the same throughout lifespan, which was a reasonable assumption at the time. Multiple studies have later challenged this assumption, showing that ~50% of mutations accumulate before maturity (*Dollé et al., 2000*; *Finette et al., 1994*; *Giese et al., 2002*; *Horvath, 2013*), although this early life pattern of mutation accumulation is not universally observed (*Welch et al., 2012*; *Blokzijl et al., 2016*; *Osorio et al., 2018*). The subsequent deceleration of mutation accumulation is now explained by the considerable slowdown of stem cell division rate upon maturation (*Bowie et al., 2006*; *DeGregori, 2013*; *Rozhok and DeGregori,*

*2015*; *Sidorov et al., 2009*). This departure from Armitage and Doll's original assumptions has been proposed to interfere with the MMC performance (*DeGregori, 2013*; *Rozhok and DeGregori, 2015*), but this has never been tested and is not considered by many cancer researchers.

None of the above-mentioned early founders of the theory had an understanding of the diversity of cancer driver mutations at the time. Armitage and Doll therefore assumed a process driven by approximately six sequentially acquired mutations, based on cancer incidence increasing as the sixth power of age, and mentioned that many cancers behave analogously, with a supposedly similar underlying process of carcinogenesis. Further research, however, demonstrated that cancers are driven by very different mutations and require very different numbers of drivers, and yet these cancers still show very similar temporal distribution with age. Later, the theory was solidified by Peter Nowell who developed the concept of clonal selection during carcinogenesis (*Nowell, 1976*) and put the model within its current general framework. Some fundamental problems, however, were already noticed by early theorists, when Fisher and Hollomon mentioned that childhood cancers drastically deviate from the observed age-dependent pattern and are difficult to explain (*Fisher and Hollomon, 1951*). An additional complication that early theorists were unaware of is the fact that different cancers originate from very different stem cell systems, with different lifetime stem cell numbers and cell division profiles. More recent models of age-dependent cancer incidence do operate with the stem/progenitor cell paradigm of cancer origins (*Calabrese and Shibata, 2010*; *Gerstung and Beerenwinkel, 2010*; *Michor et al., 2004*; *Beerenwinkel et al., 2007*; *McFarland et al., 2014*). However, the analytical models used so far have not been tested for their ability to explain age-dependent incidence for cancers driven by different numbers of mutations and originating in stem cell pools of varying sizes. Some models are limited to the single-cell perspective (*Calabrese and Shibata, 2010*) and do not account for the effect of clonal expansions in the successive accumulation of mutations. Other models are limited to modeling a typical (generic) cancer process or to a single cancer. Models comparing and unifying the principles of cancer progression of multiple cancer types and across multiple species are lacking. Critically, the current MMC lacks consideration of fundamental evolutionary processes shaping the evolution of animal life history traits (such as longevity, body size, and reproductive strategies) and, as such and despite multiple claims otherwise, has so far not been placed within the framework of evolutionary theory.

While different cancers vary significantly in lifetime risk and total incidence, most cancers have very similar temporal incidence dynamics, demonstrating approximately the same fractional increases in incidence with age. Given the vastly different etiologies and tissues of origin, this temporal incidence similarity poses a great problem to the current multi-stage model. Cancers driven by one mutation, such as chronic myeloid leukemia (CML) in chronic phase, exhibit age-dependent kinetics of incidence that are quite similar to cancers that require many more mutations, including leukemias like acute myeloid leukemia (AML) and chronic lymphoblastic leukemia (CLL) thought to also originate in hematopoietic stem cell (HSC) pools and colon cancers which originate in highly fragmented stem cell pools (*Figure 1A–C*). Moreover, hematopoietic clones driven by a single driver mutation, also thought to originate in HSC pools, have also been shown to increase with age, following a very similar incidence increase as mutationally more complex cancers (*Figure 1D*). Solutions have been proposed to explain the exponential age-dependent increase of single-mutation cancers (*Michor et al., 2006*), however the tuning of the incidence of multiple cancers with various numbers of driver mutations and originating from vastly different stem cell pools has not yet been explained. This lack of explanation is further aggravated when another dimension is added – the incidence of different cancers across different species. As in humans, cancer incidence closely follows the physiological aging curve in other animals (*Albuquerque et al., 2018*). Incidence thus scales to lifespan, but not to body size, a problem demonstrating significant deviations of cancer incidence from cell numbers and known as Peto's paradox (*Peto et al., 1975*).

We have previously proposed that explaining the above mentioned problems of the MMC requires incorporation of evolutionary theory (*DeGregori, 2013*; *Rozhok and DeGregori, 2015*; *DeGregori, 2011*; *Rozhok and DeGregori, 2016*). Somatic selection in tissues, just like selection in populations, depends on tissue microenvironment. The long co-evolution of stem cells and tissue regulatory processes should have adapted stem cells to the well-regulated microenvironments of a fit body, which should make the normal cells highly fit and should promote stabilizing selection and purifying selection against most phenotype-altering mutations. Aging disrupts tissue ecosystems,

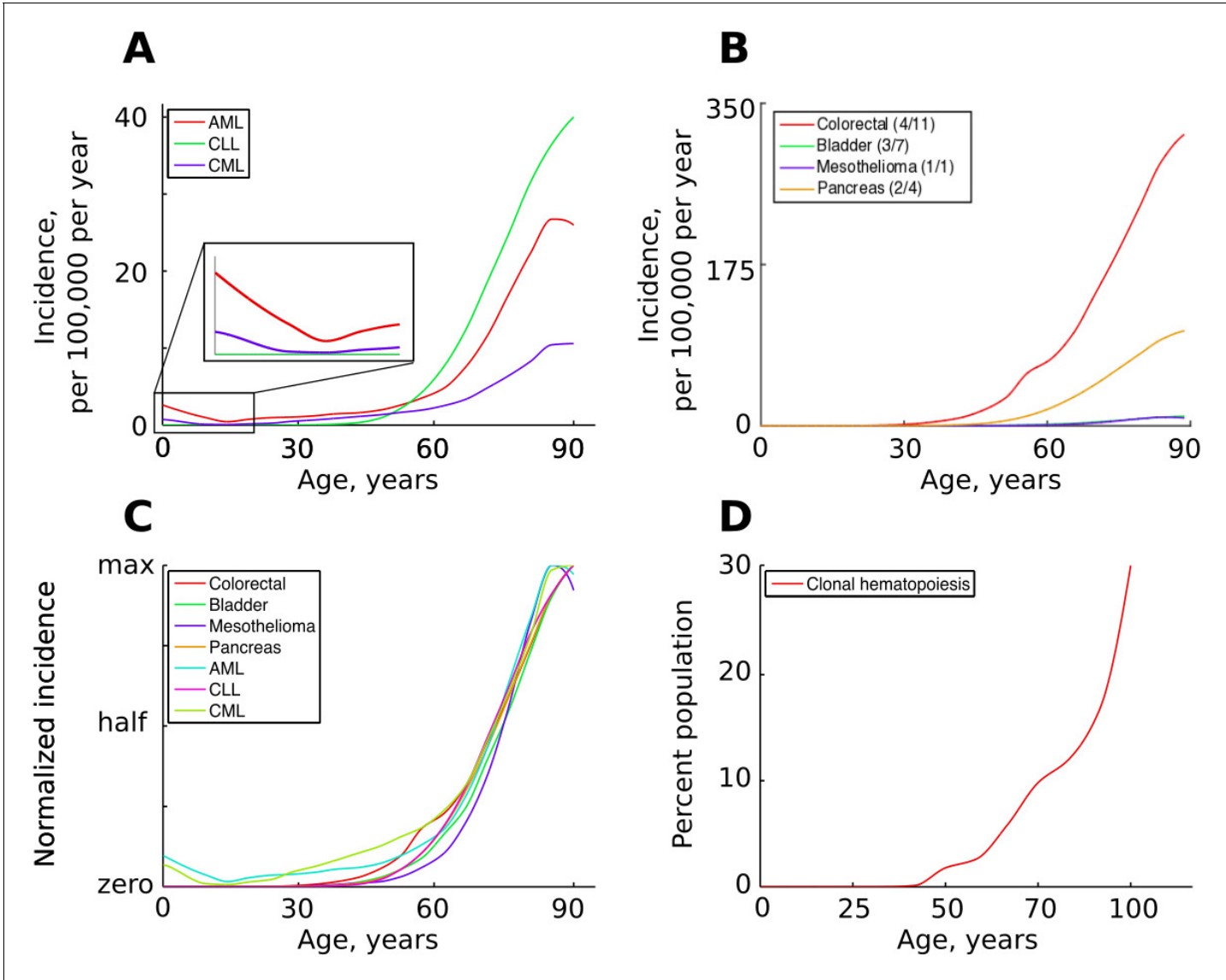

**Figure 1.** The incidence of some human cancers and clonal hematopoiesis. (**A**) The age distribution of the three most common types of leukemia: AML – acute myeloid leukemia, CLL – chronic lymphocytic leukemia, CML – chronic myeloid leukemia (data of National Cancer Institute, www.seer.cancer.gov). (**B**). The incidence of four other cancers (data of National Cancer Institute, www.seer.cancer.gov); the first number in the brackets indicates the average number of predicted driver mutations in known cancer genes and the second is the average number of predicted driver mutations in all protein coding genes according to *Martincorena et al. (2017)*. The four cancers were chosen for their variability in the predicted numbers of driver mutations. (**C**) The incidence of pooled cancers from panels A and B normalized by dividing each data point by the corresponding cancer's maximum incidence (removing scale and preserving shape). (**D**) Percent of the human population characterized by detected clonal hematopoiesis according to *Jaiswal et al. (2014)*; clonal hematopoiesis is detected as an increased variant allele frequency and is thought to be driven by a single mutation.
DOI: https://doi.org/10.7554/eLife.39950.002

and like environmental change for organismal populations, should promote positive selection for random mutants, including oncogenic mutations, that are adaptive to the new microenvironments.

In the present study, we employ the power of the Monte Carlo method to investigate the effect of various factors, such as mutation rate, cell division profiles, the impact of somatic mutations on somatic cell fitness, and the effect of aging on somatic selection and multi-stage carcinogenesis in order to test the modern MMC model and determine if age-dependent alterations in selection are necessary to explain cancer incidence patterns. We assume in this study that the character of age-dependent clonal evolution and clonal expansions is tightly linked with age-dependent cancer

incidence, which has been shown for the hematopoietic system (*McKerrell et al., 2015*; *Genovese et al., 2014*; *Jaiswal et al., 2014*; *Xie et al., 2014*). Our results demonstrate that the MMC model holds generally and is robust against departures from mutation accumulation linearity required by the original Armitage-Doll model. The MMC thus reproduces the general exponential shape of cancer incidence and clonal dynamics regardless of whether stem cell divisions remain stable or change dramatically during body maturation. However, differences in the required number of driver mutations, mutation rate and driver strength cause the MMC to fail to reproduce the expected temporal correlation in the incidence of different cancers. The current MMC is also incapable of explaining the incidence of childhood leukemia. We demonstrate that differential aging-dependent somatic selection improves the performance of the MMC and partially resolves these problems. We propose additionally a theoretical model for how reproductive success and the evolution of life history traits shape the evolution of species-specific and cancer etiology-specific tumor suppressor mechanisms. We argue that the latter mechanism should complement aging-dependent alterations in somatic selection and resolve the observed problems in MMC performance. We generalize the MMC, aging-dependent general shifts in tissue-level tumor suppressive processes, and the evolution of group-specific cellular mechanisms into a theoretical framework that we propose to better explain cancer incidence both across multiple cancer types and in multiple species.

## Quick guide to model

Our model operates with a simulated pool of cells of varying dynamic or stable sizes competing for niche space as one effective population. This design most closely replicates clonal processes in the human HSC system of the bone marrow. Multiple studies demonstrate that HSCs are effectively one population of cells that divide, differentiate and compete for a limited bone marrow niche space (*Abkowitz et al., 1996*; *Abkowitz et al., 2000*; *Catlin et al., 2011*). Their spatial segregation in different bones is compensated by the HSC's well-established proclivity to migrate (*Wright et al., 2001*), making their competition to a large extent uniform across the body over time. HSCs represent a good population for modeling somatic evolution, as most leukemias are believed to initiate in the HSC compartment (*Fialkow et al., 1967*; *Jan et al., 2012*; *Kikushige et al., 2011*; *Miyamoto et al., 2000*; *Shlush et al., 2014*). The chart of modeled events is shown in *Figure 2A* and represent a tree of possible scenarios and cell fate decisions during each update of the model. Updates are 'weekly' and continue for a total of 100 years of the simulated lifespan. The size of the simulated SC pool increases early in life and reaches its adult size by 18–20 years of age (*Figure 2B* upper chart). Cell division rates change in a similar age-dependent manner (*Figure 2B* lower chart) during body growth and maturation (*Bowie et al., 2006*; *Sidorov et al., 2009*), or division rates are kept stable (for experimental purposes). While the parameters used were derived from estimates for human HSC, *relative* changes in pool size and division rates should be similar for other tissues and other animals, given the rapid increase in body size from fetus to adult.

We simulated clonal dynamics under two different paradigms: a) driver mutations have a constant age-independent driving potential and always increase cellular somatic fitness, and b) the selective advantage conferred to cells by driver mutations depends on age, being more positive late in life and negative or neutral during the pre- and reproductive periods, as shown in *Figure 2C*. Each chart in *Figure 2C* from top to bottom shows three ways of manipulating age-dependent shifting selection: (a) selection curve shift along the fitness effect axis, varying the relative strength of early negative and late positive selection for somatic cell mutants, as this ratio, if the modeled selection shift exists in nature, is unknown; (b) varying the absolute general magnitude of selection, which is unknown as well; (c) age distribution of the selection shift, emulating differences in evolved lifespans.

We tracked the proportions of clones carrying different numbers of oncogenic mutations (1 through 4) under the influence of several primary factors:

(a) cell division speed and profile. HSC and other tissue stem cells are known to become much more quiescent after body maturation (*Bowie et al., 2006*; *Sidorov et al., 2009*), resulting in 40–50% of mutations accumulating before maturity (*Dollé et al., 2000*; *Finette et al., 1994*; *Giese et al., 2002*; *Horvath, 2013*) and thus departing from the assumption of linear mutation accumulation with age made by Armitage and Doll (*Armitage and Doll, 1954*) (but see also *Welch et al., 2012*; *Osorio et al., 2018*; *Blokzijl et al., 2016*). It has been proposed that such a departure should impact the timing of cancers (*DeGregori, 2013*; *Rozhok and DeGregori, 2015*).

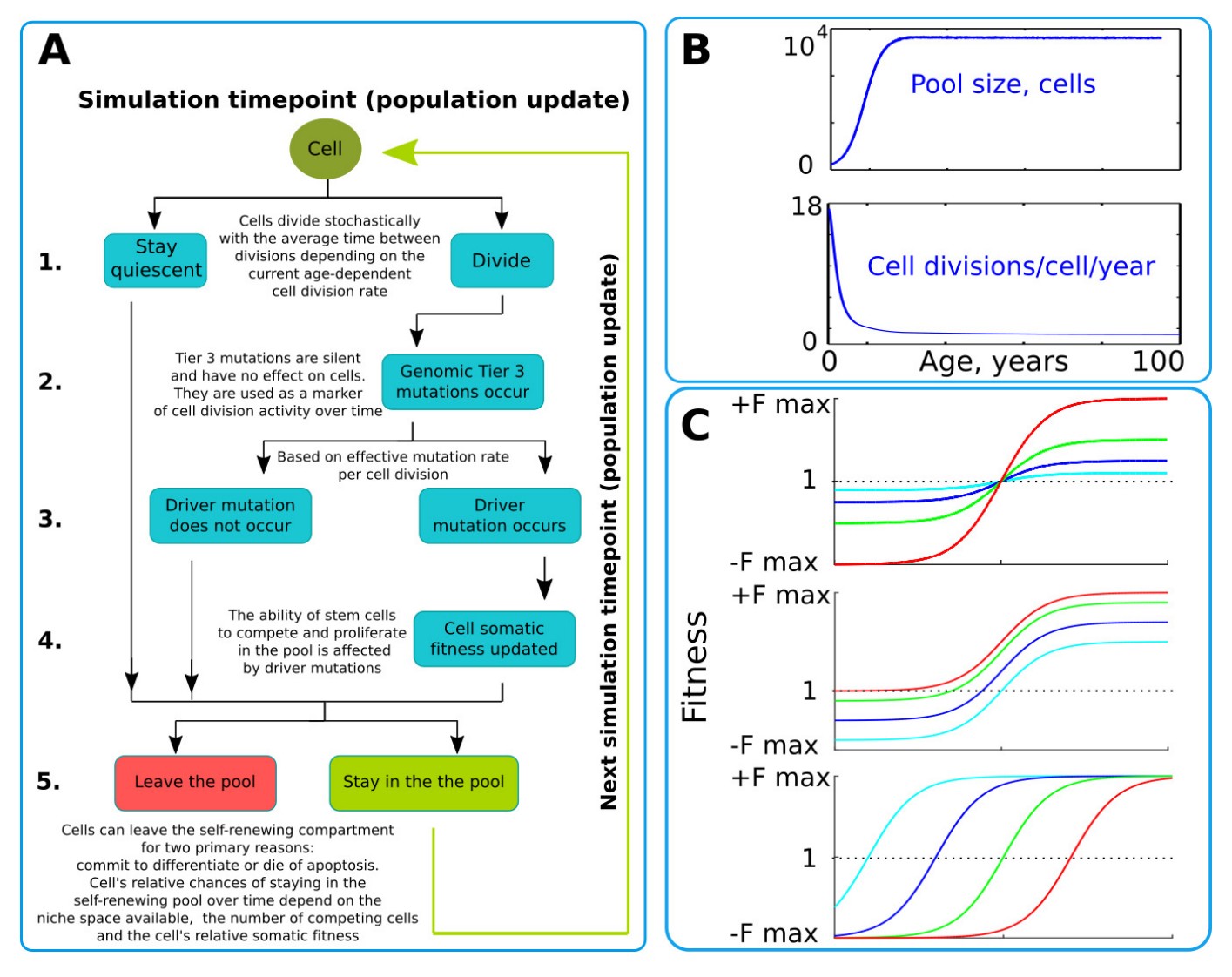

**Figure 2.** The simulation scheme and key parameters. (**A**) A tree of cell processes and cell fate decisions made by any given single cell within one simulated model update. The indicated steps are used as guideposts in the Matlab code provided in Supplementary Materials. (**B**). Non-linear age-dependent changes in the simulated SC pool size (upper chart, based on *Abkowitz et al., 1996* and *Abkowitz et al., 2002*) and cell division rate (lower chart, based on *Bowie et al., 2006* and *Sidorov et al., 2009*); the initial number of cells is 300 in simulations where the adult pool size is 10,000 cells, and proportionally larger in simulations with larger adult pool sizes. (**C**). Age-dependent shift in selection (somatic fitness effects of somatic driver mutations); somatic selection is explored within a range of general selection strengths (upper chart), a range of the ratio of the strength of early and late-life selection, and a range age distribution of selection directionality and strength, assumed in the MMC-DS model to be a function of physiological aging (reflecting evolved programs that determine longevity). The X-axis represents age from 0 to 100 years, as shown in panel B. The Y-axis represents a range of driver mutation fitness effects from maximum negative (-Fmax) through 1 (fitness equal to non-mutant cells) to maximum positive fitness (+Fmax). The top chart shows alteration in the general strength of selection (light blue:±Fmax = ±0.05%, dark blue:±Fmax = ±0.125%, green:±Fmax = ±0.25%, red:±Fmax = ±0.5%). The middle chart shows the simulated range of the relative strength of early-life negative to late-life positive selection (light blue: -Fmax/+Fmax = −0.5%/+0.5%, dark blue: -Fmax/+Fmax = −0.3%/+0.7%, green: -Fmax/+Fmax = −0.1%/+0.9%, red: -Fmax/+Fmax = 0%/+1%). The bottom chart demonstrates age-dependent selection shifts imposed by different aging profiles (age of selection sign switch in years: light blue – 10, dark blue – 30, green – 50, red – 70 years;±Fmax = ±0.5%).

DOI: https://doi.org/10.7554/eLife.39950.003

The following figure supplement is available for figure 2:

**Figure supplement 1.** An example of age-dependent clonal dynamics generated by the model.
DOI: https://doi.org/10.7554/eLife.39950.004

We simulate both a dynamic cell division profile (resulting in more mutations occurring early in life) and a stable cell division profile (which would lead to more linear mutation accumulation with age).

(b) a range of mutation rates, in order to test the MMC model behavior in tissues with presumably differing mutation loads. Noteworthy, functional mutations in the model incorporate all somatically-heritable phenotypic changes in the cell, not just nucleotide substitutions. Therefore, the simulated mutation rate does not have to match the rates observed for DNA mutations.

(c) the effect of pool size in order to explore the MMC behavior relative to body mass increase (to address Peto's paradox). Estimates for the pool size for HSC range from about 10,000 to several hundred thousand in humans (*Abkowitz et al., 2002*; *Lee-Six et al., 2018*). Notably, we are modeling pool sizes even more generally here, to encompass different animals;

(d) the strength (selective advantage) of driver mutations, which varies for different natural oncogenic mutations. We simulate two different assumptions: (1) fixed fitness effects of mutations as utilized in many modeling studies and by modern MMC theorists (*Bozic et al., 2010*; *Tomasetti et al., 2015*; *Vogelstein et al., 2013*); and (2) age-dependent varying strength of selection acting on oncogenic mutations, which is a theoretical extension of the MMC model with age-dependent Differential Selection, so we designate it as MMC-DS. As we have argued earlier, one of the central paradigms in evolutionary theory posits that selective advantage is not a fixed attribute of genetic change, but is a dynamic property arising at the interface of the resulting phenotype and environment (*Rozhok and DeGregori, 2015*). Tissue microenvironment, and thus the signaling that controls stem cell fate decisions, change in an aging-dependent manner and thus should impact somatic selection differentially as a function of physiological aging. Additional dynamical patterns of changing fitness effects of mutations are certainly possible (such as reversal of positive selection), but those will not be explored here. Cell intrinsic processes that lead to cell aging should also promote stronger positive selection processes by broadening the distribution of individual cell somatic fitness driven by variations in the accumulation of cellular damage among cells. In combination with microenvironmental changes, these processes should lead to generally increased rates of directional selection in large stem cell pools, such as for hematopoiesis.

## Results

### Cell division profiles

We first tested if early rapid cell division, unaccounted for by Armitage and Doll (*Armitage and Doll, 1954*) and in most modern modeling studies, alters the pattern of somatic evolution. We tested the MMC and MMC-DS behavior under a stable cell division rate over lifetime and one that slows down post-maturation as shown in *Figure 2B* (lower chart). The dynamics of clones resulting from accumulating a series of driver mutations can be seen in *Figure 3A*. The chart shows the general age-dependent size of the total simulated cell pool (grey curve), as well as the dynamics of clones (in absolute cell numbers), containing 1, 2, 3 or 4 somatic driver mutations (colored lines). As we argue in the Materials and methods *Model architecture* section, the exact size of the adult HSC pool is not critical for the model, as long as the pool is large enough to minimize drift. Each next mutation either improves the affected cell's somatic fitness (the standard MMC assumption) or has a varied age-dependent effect on cellular somatic fitness as shown in *Figure 2C* (the MMC-DS assumption). Plots in *Figure 3A* show simulation results for clones containing 1, 2, 3 or four driver mutations, plotting clonal representation independent of subsequent driver mutation accumulation (e.g. clonal expansions with two drivers will include sub-clones with additional driver events). The clonal dynamics presented here and subsequently are averaged dynamics of 50 repeated runs. An example showing the stochasticity and dynamics of all the 50 runs is shown in *Figure 2—figure supplement 1*. The results shown in *Figure 3A* demonstrate that non-linear changes in cell division rates early in life do not have a significant influence on the model's capability to replicate an exponential succession of clones driven by accumulating a series of driver mutations. *Figure 3A* also shows that clones with different numbers of drivers have different timing of expansions. We can therefore conclude that even with rapidly decelerating stem cell division rates causing early accumulation of mutations, under select assumptions for numbers of required driver mutations and mutation rate, the current MMC model is still capable of replicating higher late-life rates of somatic evolution, perhaps due to the waiting time necessary for initiated cells to expand into larger clones. MMC-DS also can

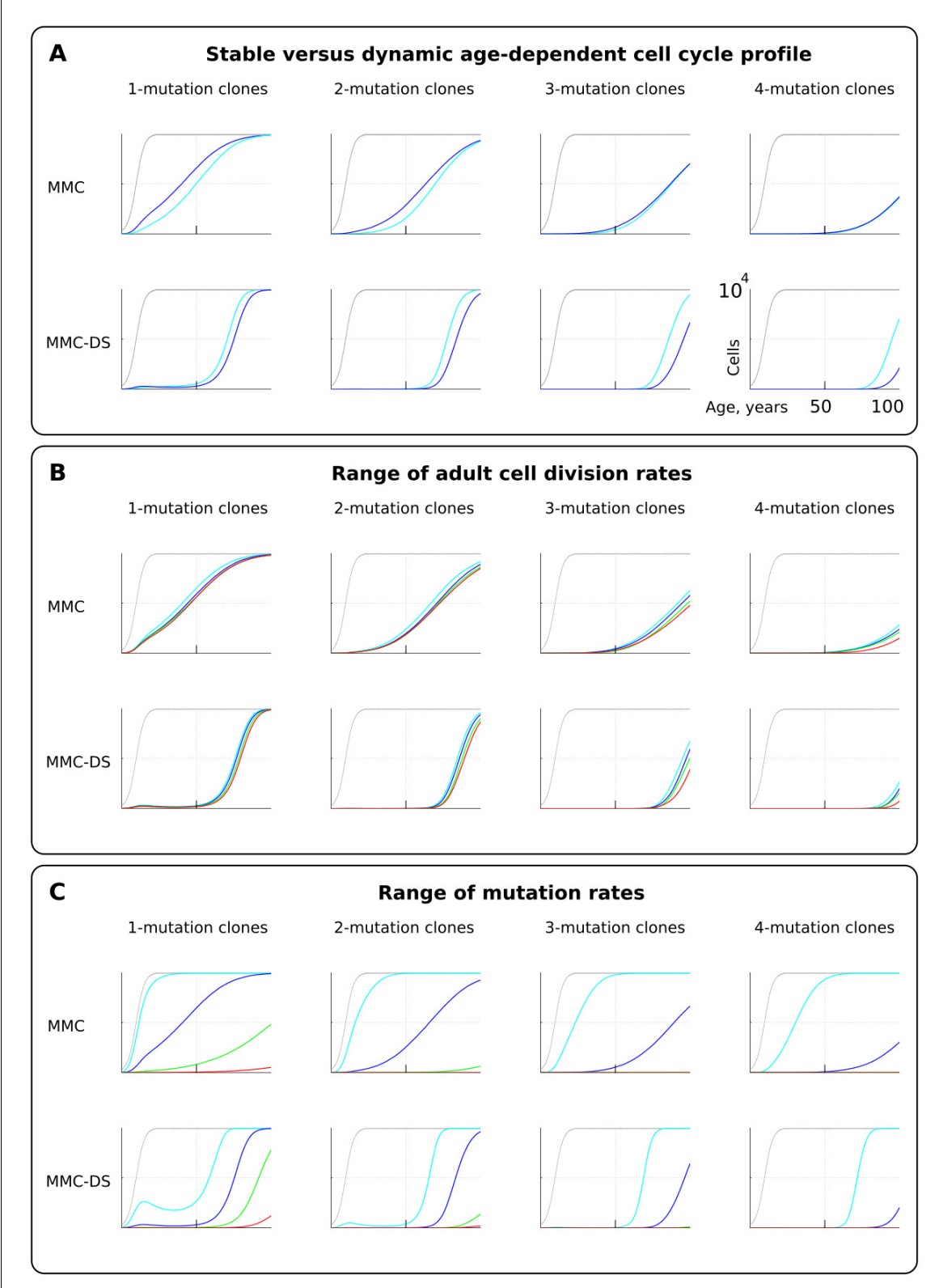

**Figure 3.** The effect of cell division profile, adult cell division rate and mutation rate on clonal dynamics of cells carrying 1, 2, 3 or four somatic driver mutations. Y-axis in all charts ranges zero to 10,000 cells (linear scale); X-axis in all charts ranges zero to 100 years of age (linear scale).(**A**). The effect of dynamic (dark blue; the curve of cell division rates as shown in *Figure 2B*, lower chart) versus stable (light blue; cells division rate is stable at one division ~20 weeks throughout lifespan) cell division profiles. Grey curve in all charts of this type represents the size of the total SC pool. Other
*Figure 3 continued on next page*

*Figure 3 continued*

parameters as listed in *Standard parameter sets, Supplementary Materials*. (B). The effect of adult cell division rate under the dynamic age-dependent cell division profile (as shown in *Figure 2B*, lower chart). Color coding: light blue – one division per cell in ~70, dark blue – one in ~60, green – one in ~50, and red – one division ~40 weeks. Other parameters as listed in *Standard parameter sets, Supplementary Materials*. (C) The effect of mutation rate (MR). Color coding: light blue - MR = $10^{-2}$, dark blue – MR = $10^{-3}$, green – MR = $10^{-4}$, and red – MR = $10^{-5}$. Other parameters as listed in *Standard parameter sets, Supplementary Materials*. Notice that MR is not the usually understood mutation rate per cell division per base pair, but is the probability of acquiring (per cell division) any genetic/epigenetic change that confers the incipient cell a change in somatic fitness.

DOI: https://doi.org/10.7554/eLife.39950.005

The following figure supplements are available for figure 3:

**Figure supplement 1.** Results of statistical comparisons related to testing the effect of dynamic age-dependent cell division profiles.

DOI: https://doi.org/10.7554/eLife.39950.006

**Figure supplement 2.** Results of statistical comparisons related to testing the effect of adult cell division rates.

DOI: https://doi.org/10.7554/eLife.39950.007

**Figure supplement 3.** Results of statistical comparisons related to testing the effect of mutation rates.

DOI: https://doi.org/10.7554/eLife.39950.008

reproduce late-life increases in somatic evolution rates due to the stronger late-life positive selection. The cell division profile, therefore, does not discriminate between MMC-DS and the standard MMC, suggesting that both models, so far, appear plausible. *Figure 3B* also shows that the effects of a range of adult cell division rates within the dynamic cell division profile is rather modest. As with age-dependent cell division profiles, the noticeable difference is that the MMC-DS model delays high rates of clonal evolution until the later portion of the simulated lifespan.

## Mutation rate

Different tissues have different exposures to external factors. For example, the digestive system and lungs are exposed to a variety of potential mutagens, while the skin is subjected to ultraviolet radiation. Whether counted on a per cell division basis or as a function of time, effective mutation rates should differ for different tissues. Rates also vary for different animals (*Lynch, 2010*). Higher mutation rates should accelerate somatic evolution by increasing phenotypic variability. We therefore further explored the effect of mutation rate, which is shown in *Figure 3C*. Expectedly, both MMC and MMC-DS are sensitive to mutation rate. However, we observe clear differences in the predictions by MMC and MMC-DS. The MMC-DS model, unlike MMC, prevents all clones from expanding through youth (the period of likely reproduction). As in the tests of cell division rate effects (*Figure 3A,B*), both models are sensitive to the number of driver mutations, with more drivers requiring more time to accumulate and expand the recipient clones. Still, the number of drivers has much less of an effect within the MMC-DS model, with expansion kinetics delayed until and compressed in old age.

## Fitness advantage conferred by driver mutations

For simplicity and as a demonstration of the basic principle, we set the fitness advantage conferred to cells by driver mutations to be the same for all drivers. Clones having more mutations are thus more aggressive in their expansions. This assumption reasonably replicates the basic architecture of the modern MMC. However, we know that real driver mutations vary considerably in their effects on cells. For example, the translocation generating BCR-ABL can drive a myeloproliferative disease (CML) apparently without other driver mutations (*Mulligan et al., 2008*). Other mutations require many cooperating events to make the recipient clone somatically aggressive (*Martincorena et al., 2017*). We therefore further tested the effect of different strengths of driver mutations, shown in *Figure 4A*. As intuitively anticipated, a greater fitness advantage accelerates somatic evolution rates in both models (*Figure 4A*; red is the greatest fitness advantage). However, as with mutation rate, MMC-DS, unlike MMC, delays all clonal expansions until old age (independent of driver mutation numbers). In a sense, the fitness advantage conferred by driver mutations acts on somatic evolution similarly to mutation rate – higher values accelerate the emergence and expansion of clones. While MMC has only one parameter for mutation fitness effects, MMC-DS has two: a) a general effect that impacts both the strength of negative and positive selection, and b) the ratio of negative to positive selection shown in *Figure 4A* as MMC-DS(a) and MMC-DS(b), correspondingly. These two parameters are convenient ways to test the different aspects of aging-dependent selection shifts. The

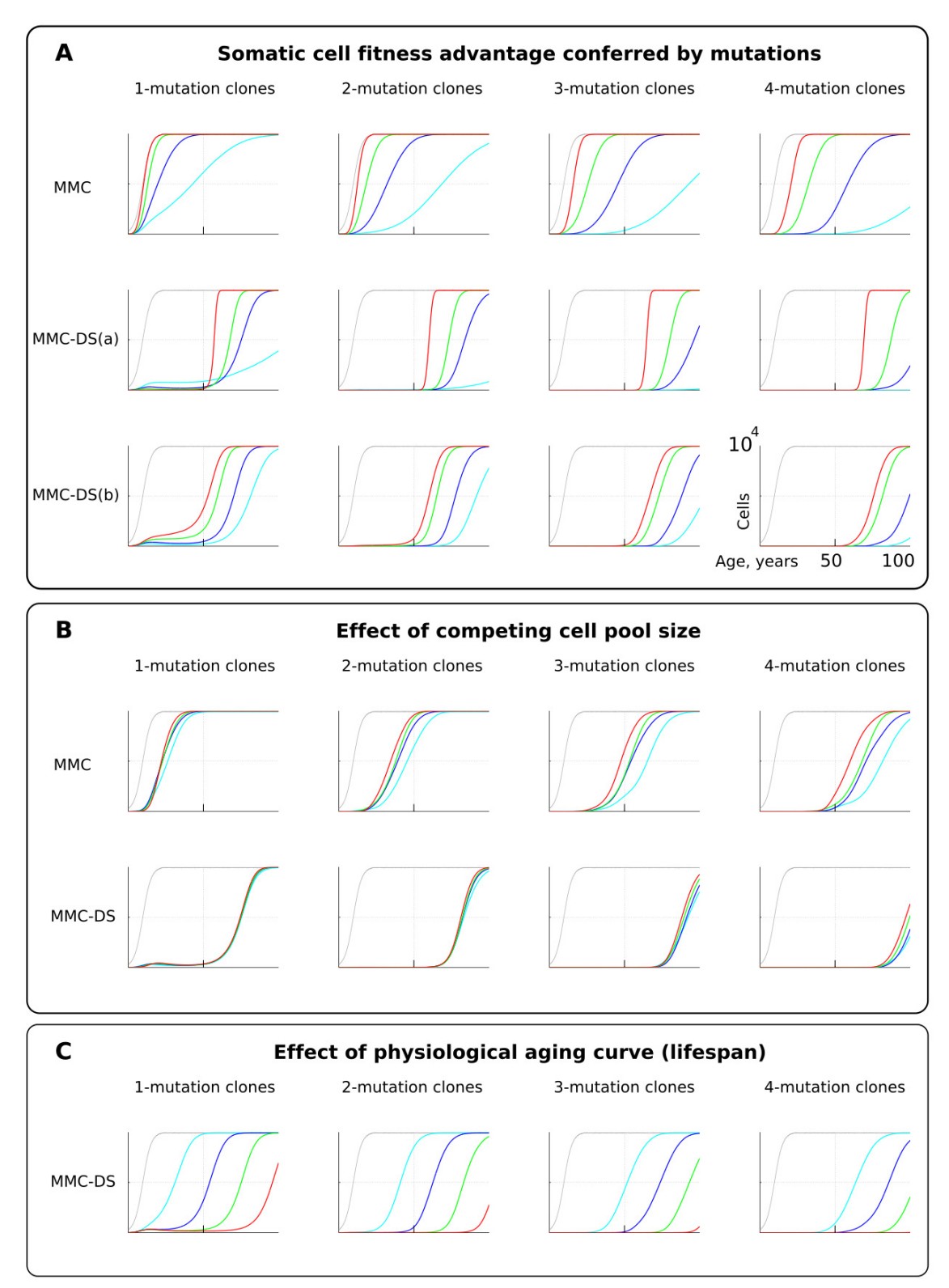

**Figure 4.** The influence of mutation somatic fitness effects, pool size, and physiological aging. Y-axis in all charts except panel B ranges zero to 10,000 cells (linear scale); X-axis in all charts ranges zero to 100 years of age (linear scale).(**A**). The effect of alterations in somatic cell fitness conferred by driver mutations. Color coding for MMC: light blue – fitness advantage (FA) =+0.1%, dark blue – FA =+0.25%, green – FA =+0.5%, and red – FA =+1%. MMC-DS(a) explores a range of the general strength of differential selection corresponding to and color-matched to *Figure 2C*, upper chart (absolute

*Figure 4 continued on next page*

*Figure 4 continued*

selection strength values are described in *Figure 2C*). MMC-DS(b) explores a range of early to late life selection strength ratios corresponding and color-matched to *Figure 2C*, middle chart (absolute values are described in *Figure 2C*). Other parameters as listed in *Standard parameter sets, Supplementary Materials*. (B). The effect of adult pool (AP) size. Color coding: light blue – AP = 10,000 cells, dark blue – AP = 25,000 cells, green – AP = 50,000 cells, and red – AP = 100,000 cells. Starting pool size in all conditions is 300 cells. The Y-axis is relative and represents percent of AP (AP = 100%, maximum size delineated by the black curve). Other parameters as listed in *Standard parameter sets, Supplementary Materials*. (C). The effects of physiological aging in the MMC-DS model. The corresponding and color matched aging curve profiles (aging mirrors the shifts in somatic selection in MMC-DS) are shown in *Figure 2C*, lower chart. Other parameters as listed in *Standard parameter sets, Supplementary Materials*.

DOI: https://doi.org/10.7554/eLife.39950.009

The following figure supplements are available for figure 4:

**Figure supplement 1.** Results of statistical comparisons related to testing the effect of somatic fitness effects conferred by driver mutations.

DOI: https://doi.org/10.7554/eLife.39950.010

**Figure supplement 2.** Results of statistical comparisons related to testing the effect of adult cell pool size.

DOI: https://doi.org/10.7554/eLife.39950.011

**Figure supplement 3.** Results of statistical comparisons related to testing the effect of physiological aging curve (a proxy for lifespan).

DOI: https://doi.org/10.7554/eLife.39950.012

MMC-DS(b) modeling (*Figure 4A*) demonstrates that it is not necessary for early life selection acting on somatic mutants to be negative. Even in the absence of negative selection (red lines, the early life fitness value is the same as for non-mutant cells), the age-dependent character of clonal expansions predicted by MMC-DS holds. This result demonstrates the general idea that whatever the effects of driver mutations early and late in life, MMC-DS demonstrates clonal expansions approximating known cancer incidence if the driver potential of mutations early in life is sufficiently reduced relative to old ages.

## Pool size

One of the most interesting challenges to the current MMC discussed in a number of papers is the question why larger animals do not suffer a proportionally higher risk of cancer, known as Peto's paradox (*Dang, 2015*; *Ducasse et al., 2015*; *Tollis et al., 2017*). Larger animals, all other traits equal, clearly present a larger target size (cell division numbers) for cancer mutations, increasing the likelihood of their occurrence. However, incipient oncogenic cells and clones must compete with a larger pool of cells in order to develop into a life-threatening tumor (a tumor of 1 cm in diameter will likely kill a mouse, while hardly posing a tangible threat for a whale). Absolute cancer cell numbers, independent of the proportion, are also important, as the number defines the likelihood of cancer progression through successively accumulated mutations, and thus it defines the likelihood of developing advanced cancers. However, the ultimate mortality risk also depends on the proportion of such cells relative to the affected tissue. Thus, the higher opportunity for mutations to occur in larger bodies is counteracted by the longer path to a life-threatening cancer due to increased suppression and larger tolerance of tumors exerted by larger tissues. We therefore further tested if altering the simulated cell pool size will affect the proportions of mutant clones relative to pool size. *Figure 4B* shows that the mentioned factors effectively counteract each other – increasing pool size does not increase the relative presence of mutant clones in it, and this relationship largely holds for both MMC and MMC-DS. However, MMC appears more sensitive to stem cell pool size (which should increase for larger body size). This test demonstrates the effect of two opposing forces acting on the frequencies of somatic mutant clones. A larger body provides more cells and cell divisions, and thus a greater target for mutations. However, a mutant malignant cell will need a longer time and a larger expansion to become a threat. Under the MMC-DS model, the process is further affected by the fact that mutants emerging early in life are universally (mostly) purged from the pool.

## Aging curve

The aging curve as a factor affecting the rates of somatic evolution is a trait unique to the MMC-DS paradigm. We tested the response of clonal dynamics to aging profiles. This was done by shifting the curve that determines age-dependent selection acting on mutation clones as shown in *Figure 2C*, lower chart. Such shifts effectively imitate different lifespans under the assumption that age-dependent somatic selection mirrors the aging curve (*Rozhok and DeGregori, 2016*). This can

be extrapolated either on the lifespan differences between various animal species or various life-styles or genetic factors among humans. *Figure 4C* shows, as expected, that earlier aging that produces earlier onset of positive selection on mutant cells leads to corresponding earlier expansions of mutant clones accumulating a series of driver mutations.

## Early life clonal expansions

As can be seen in results presented in *Figure 3* and *Figure 4*, especially for the dynamics of clones with one driver mutation, the MMC-DS model appears to differ from MMC in clonal dynamics very early in the simulated lifespan, with the early appearance and disappearance of driver-containing clones in MMC-DS. We therefore further simulated clonal evolution with a larger sample of simulated individual lives. *Figure 5A* shows a total of 25,000 individual runs (not averaged as before) under the MMC assumption. All clones demonstrate uninterrupted increases in clonal size until peaking. However, the dynamics of clones under the MMC-DS assumptions notably differ. *Figure 5B* shows that under the MMC-DS assumption of shifting selection, many clones demonstrate a minor peak very early in life, but are later suppressed until the second half of the simulated lifespan. This difference holds under a wide range of other parameters, apparently because of the negative selection that is needed to suppress mutant clones during the first half of lifespan. The reason why very early clones are capable of forming small peaks is the larger presence of random drift imposed by the smaller pool size relative to the adult pool, interacting with faster cell division rates (generating more mutations). Smaller population sizes promote drift and weaken selection. If selection acting on driver mutations is always positive, such an early peak is unlikely to form. As we have previously shown (*Rozhok et al., 2016*), such clonal dynamics in HSC pools might provide an explanation for the elevated rates of leukemia in early childhood. As can be seen in *Figure 5B*, even clones that have accumulated four drivers show such a peak, with such clones becoming progressively fewer and smaller as the number of accumulated drivers increases. Childhood leukemias are quite rare relative to late-life cancers. However, these results show that replication of the full pattern of leukemia incidence requires the MMC-DS model assumption of differential selection, whereby early portions of lifespan are characterized by a general suppression of somatic evolution by means of purifying selection.

## Discussion

Our results indicate that the general principle of successive cell transformations proposed by Nordling, Armitage and Doll (*Nordling, 1953*; *Armitage and Doll, 1954*) and later developed into the modern Multi-Stage Model of Carcinogenesis (MMC) recapitulates the late-life exponential increases in the rates of somatic evolution regardless of the non-linear pre-maturity shifts in stem cell division rates. The model, however, requires the incorporation of differential aging-dependent somatic selection acting on somatic cell variants, a principle earlier proposed as a key postulate of the Adaptive Oncogenesis model (*DeGregori, 2011*), in order to universally postpone increased rates of somatic evolution to late ages, independent of the number of driver mutations and the magnitude of the somatic selective value of these mutations. This principle of aging-dependent somatic selection can potentially explain the scaling of increased cancer incidence to the species-specific lifespan, such that most cancers occur after 1.5 years for mice and 50 years for humans.

The current MMC does not explain a very puzzling phenomenon – that most cancers show strikingly similar late life patterns of incidence, despite the fact that these cancers require very different numbers of driver mutations and originate in very different sized/organized stem cell pools (whether for different tissues or in different species). Previous attempts to explain the age-dependence of multi-stage carcinogenesis have mostly used analytical modeling, with fixed effects of mutations on cellular fitness (*Calabrese and Shibata, 2010*; *Gerstung and Beerenwinkel, 2010*; *Michor et al., 2004*; *Beerenwinkel et al., 2007*; *McFarland et al., 2014*). However, fitness is a dynamic phenomenon that arises at the interface of phenotype and environment and is determined by the mode of selection acting on a phenotype, the latter being typically defined by environment (*Wade and Kalisz, 1990*). While with analytical modeling parameters can be derived that fit the late-life curve for a particular cancer, these studies do not explain what evolutionary process/mechanism fine-tuned the occurrence of multiple cancers of vastly different etiology and harboring different numbers and spectra of mutations.

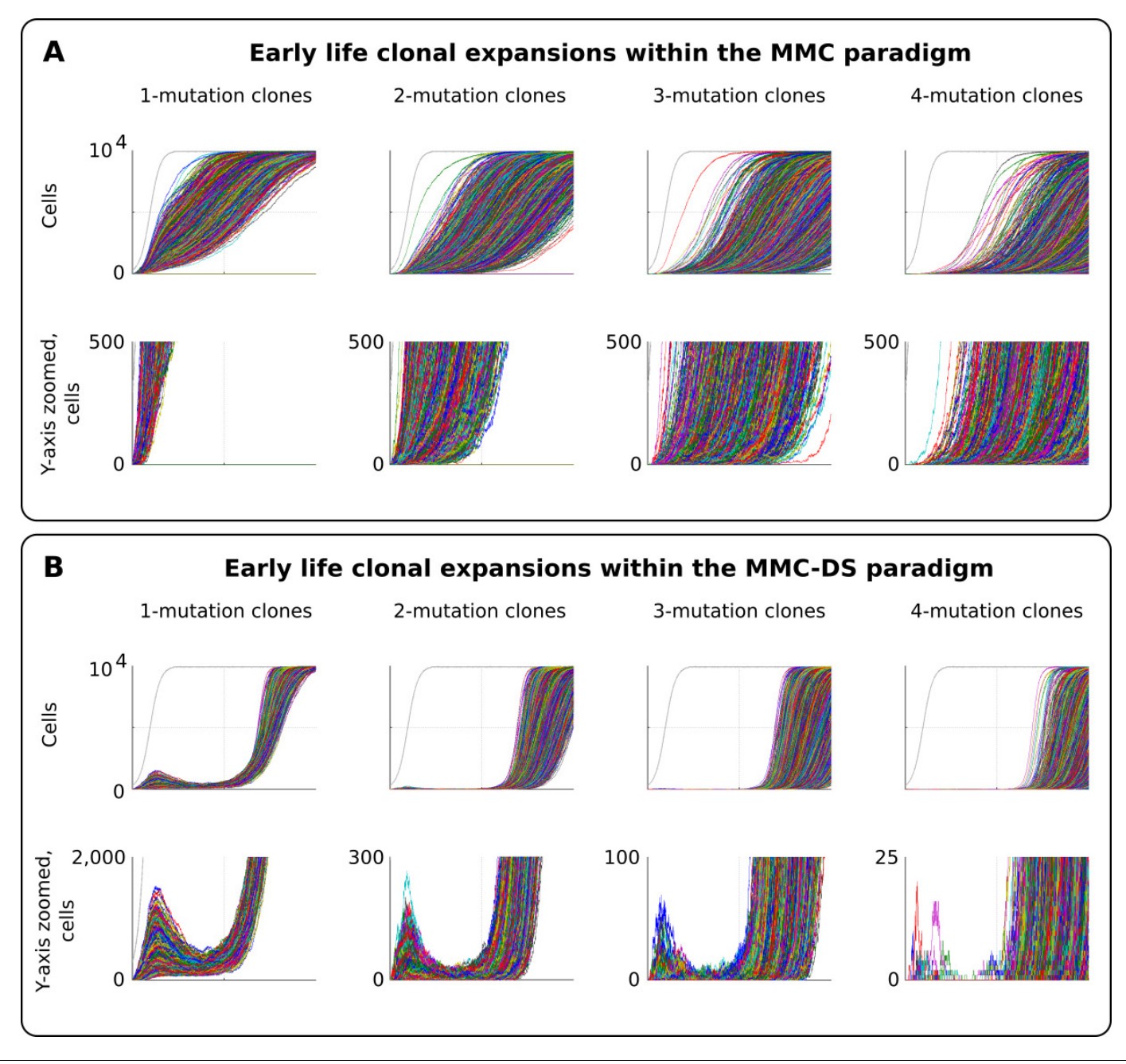

**Figure 5.** Clonal dynamics promoted by somatic driver mutations in early developing HSC pools. (**A**) The MMC model demonstrates an uninterrupted increase in mutant clone frequencies over time. *Other parameters as listed in Standard parameter set MMC, Standard parameter sets, Supplementary Materials.* (**B**). The MMC-DS model demonstrates early peak frequencies of mutant clones which are later suppressed by purifying selection acting throughout the early half of the simulated lifespan. *Other parameters as listed in Standard parameter set MMC-DS, Standard parameter sets, Supplementary Materials.*

DOI: https://doi.org/10.7554/eLife.39950.013

The MMC-DS model also demonstrates early life increases in the rates of somatic evolution, mirroring the risk of early childhood cancers, which does not arise from the standard MMC model. As we have argued before (*Rozhok et al., 2016*), these early-life processes should be driven by drift due to the small initial pool of stem cells in a fetus and right after birth. Such domination of drift has also been shown experimentally for tissues in which stem cell compartments are fragmented into

small local populations (*Vermeulen et al., 2013*). Later suppression of somatic evolution until older ages can be explained by purifying somatic selection that acts in larger adult stem cell pools during reproductive periods of life-spans. Consistent with this logic, early childhood cancers mostly comprise malignancies originating in large stem cell systems, such as hematopoietic, while childhood carcinomas, which originate in the small fragmented epithelial stem cell compartments where drift dominates, are rare (www.seer.cancer.gov) (*Rozhok et al., 2016*). Unlike large pool SC systems, the strong presence of drift should limit positive somatic selection in fragmented epithelial SC compartments throughout lifespan. As modeled in the MMC-DS setup, oncogenic mutations that are negatively selected in youth can be positively selected late in life, most likely due to alterations in microenvironments which alter the adaptive value of mutations. Indeed, previous studies have demonstrated how the somatic fitness effects of oncogenic mutations can be dramatically affected by the age of the host (*Henry et al., 2010*; *Henry et al., 2015*; *Parikh et al., 2018*; *Vas et al., 2012*).

Nevertheless, the key question that arises from our results and within the general MMC model is what mechanism underlies the temporal coincidence of the increased cancer rates among cancers of vastly different etiology driven by different driver mutations and different numbers of driver mutations? The general MMC-DS architecture resolves the problem of early postnatal elevated rates of somatic evolution and the scaling of somatic evolution rates to lifespan, with increased rates of somatic evolution delayed until late in life. However, the general MMC-DS principle is not sufficient to overcome the problem of different numbers of driver mutations for different cancers with similar age-dependent patterns. For example, and as shown in *Figure 3A*, the incidence curve for a 1-driver cancer is significantly earlier than for a 4-driver cancer, while the same is not observed for actual cancer incidence curves (*Figure 1C*). While it is difficult to imagine what evolutionary forces could have 'tuned' the incidence of vastly discrepant cancers within the standard MMC paradigm, we propose a theoretical generalization to explain this phenomenon, which integrates the current MMC theory, the principle of aging-dependent shifts in somatic selection, and the group-specific evolution of cellular machinery that depends on the evolution of life history traits. This generalized theory can also explain the early childhood cancer incidence peaks and Peto's paradox. We postulate that the following principles should operate in shaping the general character of somatic evolution in order to resolve the current problems of the MMC:

## The principle of a mutual feedback between germline and somatic selection

Based on the above presented results, we propose that somatic selection in the body and germline selection in a population are directly linked with mutual feedback acting on each other through reproductive success as shown in *Figure 6*. Somatic selection, as the main determinant of somatic evolution in a tissue, impacts age-dependent tissue and individual fitness, and thus provides the fuel for germline selection operating on individual fitness. In this way, the effect of somatic mutations on cell somatic fitness is subject to germline selection based on an individual's fitness/health risks conferred by particular somatic mutations. Reproductive success provides a quantitative effect whereby the strength and directionality of germline selection acting on particular somatic cell alterations is determined by the overall reduction in reproductive success of an individual. As demonstrated in *Figure 6*, such cost diminishes with age proportionally to the likelihood that an individual of a particular species will die at a specific age of causes less unrelated to health (such as predation). Therefore, germline selection acts on the effects of somatic mutations on cellular physiology depending on a species-specific general age-dependent mortality profile.

Accordingly, the somatic fitness effects of different mutations should differ within a species. Principle #1 can help explain the similar age-dependent incidence curves shown in *Figure 1*, despite the fact that these cancers originate in very different sized stem cell pools and require very different numbers of driver mutations. For example, in the MMC-DS(b) setup in *Figure 4A*, the expansion of a 1-mutation clone (light blue curve in the leftmost chart) will approximate the expansion of a 4-mutation clone (green curve in the rightmost chart), if germline selection 'equalizes' the effect of the single mutation in the first case and the combined effects of the four mutations in the second based on their overall health risk, following the logic presented in *Figure 6*. Basically, germline selection acting to determine the fitness effects of each somatic mutations functions as the 'invisible hand' that pushes cancer risk to older ages where reproductive success is low.

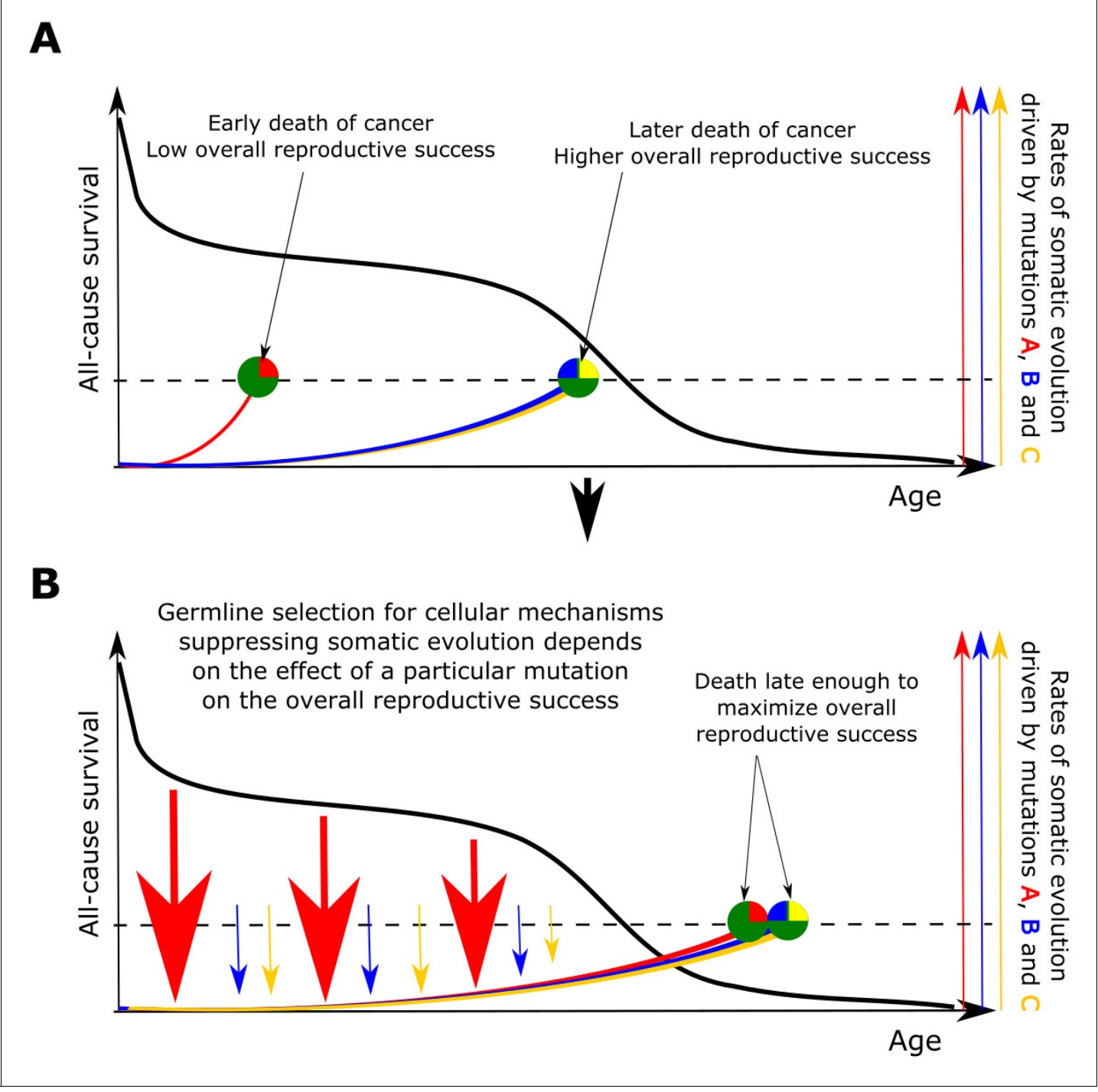

**Figure 6.** The principle proposed to underlie fundamental properties of somatic evolution in animal bodies. (**A**) The effect of somatic evolution on germline evolution. Any particular somatic mutation can range on a scale from no health risk to a threat of early death. Green circles represent individual animals; red, yellow and blue are somatic mutations. The red mutation confers high risk of carcinogenesis causing early death (dashed line represents lethal threshold from somatic evolution). The red mutation thus has a profound effect on overall individual reproductive success. The yellow and blue somatic mutations need to cooperate to cause a lethal effect, which delays somatic evolution driven by them, as shown by our modeling results and earlier postulated by Armitage and Doll (***Armitage and Doll, 1954***). Therefore, the yellow and blue mutations have lower impact on reproductive success. As reproductive success is the universal equalizer impacting organismal fitness, germline selection acts to buffer the cellular effects of the red mutation differently than it acts on the effects of the yellow and blue mutation. (**B**). As a result of germline selection, based on the impact of each mutation on reproductive success, stronger tumor suppressive mechanisms evolve to limit the negative impacts of the red mutation. This effect is quantitative in that the differential amount of germline selection acting on a particular mutation depends on the degree of decrease in the

*Figure 6 continued on next page*

*Figure 6 continued*

overall reproductive success conferred by each mutation's health risk. The resulting tumor suppression mechanisms delay cancer risk. However, such a delay cannot extend significantly beyond the ages when most of the population would be dead for other reasons, as the cost of developing tumor suppressor mechanisms is no longer compensated by the benefit of higher overall reproductive success at these later ages. Such a model implies that germline selection for tumor suppressive mechanisms is somatic mutation-specific and species-specific.

DOI: https://doi.org/10.7554/eLife.39950.014

## The evolution of life history traits impacts the feedback between germline and somatic selection

Germline selection acts on individual fitness during the period of population-specific likely survival in nature. Based on the species' ecological niche, such selection thus governs the evolution of longevity, resulting in a species-specific aging curve (*Medawar, 1952*; *Hamilton, 1966*; *Williams, 1957*; *Tuljapurkar et al., 2007*). It follows then that individual size, longevity and physiological aging profile determine the impact somatic mutations have on germline selection. For example, the contribution to mortality and individual fitness of an oncogenic mutation that requires 10 years to develop into a cancer will differ for two species with all traits equal, but different longevity. If a species does not survive in nature beyond 10 years for other reasons (such as predation), such a mutation will not impact germline selection and thus, following principle #1, germline selection will not act on cell fitness effects of such a mutation. Similarly, animals that develop defensive mechanisms against their major causes of mortality, such as increasing body mass to withstand predation, and extending thus their longevity, simultaneously increase the impact of somatic cell alterations through later ages on their overall reproductive success. For example, the risk of dying of cancer at the age of 10 for a cat will have much lower impact on the overall reproductive success (of an affected individual relative to others) than such a risk at age 10 for an elephant. In other words, the relative contribution of external mortality risks and health-related risks to the overall reduction in reproductive success at a given chronological age is different for species with different body size and longevity.

## Somatic fitness effects of similar mutations should differ in different species

Following principles #1 and #2, analogous cell phenotype-altering mutations should have different effects on cell somatic fitness in species with different life history traits as a result of evolution driven by different germline selective pressures on the effects of such somatic mutations on cell physiology. For example, an analogous cancer driver mutation in a mouse and a human should have different effects on the somatic cell fitness of the incipient cells and thus different cancer driving potential. Experimental support of this idea already exists, as freshly isolated mouse fibroblasts, for example, can be transformed into cancer-like cells in culture with only two oncogenic mutational hits, while human cells require up to six (*Rangarajan et al., 2004*).

## Embryonic and early postembryonic development provides a somatic selection-independent 'window of opportunity' for increased rates of somatic evolution driven by somatic drift

Based on the results shown in *Figure 5* and earlier published results (*Rozhok et al., 2016*; *Vermeulen et al., 2013*), even in the presence of purifying selection against somatic mutant cells, a small self-renewing cell compartment should impose somatic drift that largely buffers selection. This principle can help explain the incidence spikes of childhood cancers. However, actual cancer development in children can be promoted by other factors as well. The initially drift-driven expansions could later overcome the uniformly negative selection acting on driver mutations in youth (e.g. if the third driver becomes positively selected in the presence of the first two drivers) or under the impact of other factors, such as inflammation, that might lead to positive selection of drivers (*Greaves, 2006*). However, it should be stressed that the fact that the incidence of such cancers dwindles into early adulthood can be explained by purifying selection acting against somatic cell mutants.

Based on these principles and the mechanisms shown in *Figure 6*, we therefore argue that aging-dependent differential selection regulated at the tissue level and being likely a mechanism common across animal species should work in concert with additional inherited mechanisms of tumor suppression that are shaped by the evolution of life history traits of a particular species. These latter mechanisms may be similar in different animal groups, based on similarities in the general architecture of cell regulatory networks. However, significant differences that are specific to particular animal groups should not be surprising from this standpoint, as evolution is also a game of chance and opportunity. Evidence of such unique mechanisms already exists for multiple species (*Seluanov et al., 2018*). A notable example is a specific type of hyaluronic acid in the naked mole rat that is believed to underlie the species' exceptional cancer protection capabilities (*Tian et al., 2013*). Similarly, alterations in p53 genes in elephants represent a potential group-specific mechanism to delay tumorigenesis till older ages (*Abegglen et al., 2015*; *Sulak et al., 2016*). Based on the principle shown in *Figure 6*, tumor suppressive mechanisms tailored to limit the impact of classes of cancer driver mutations (or particular pathways altered in cancers) could be either species-specific or common across mammals or even animals. The general aging-dependent shift in somatic selection and the evolved group-specific pathway-tailored tumor suppressor mechanisms can be viewed as essentially two different dimensions of which the overall mechanism regulating tumor suppression in animals is composed.

Alongside childhood cancers, a number of cancers deviate from the general late-life pattern and occur at earlier ages. These cancers tend to be associated with infections (e.g. Burkitt's lymphoma) (*Grywalska and Rolinski, 2015*), to be very rare (e.g. testicular cancers; www.seer.cancer.gov), and/or associated with modern lifestyles and conditions (e.g. breast cancer) (*Layde et al., 1989*; *Hochberg and Noble, 2017*; *Giraudeau et al., 2018*). The incidence of cancers of such etiologies is largely consistent with the proposed principles outlined above in the sense that their causes represent factors a) for which human fitness is in conflict with that of a pathogen, b) where selection is weakened by the rare manifestation of the disease, and c) where the evolution of tumor suppression did not have enough time to counteract modern changes in lifestyle, respectively. The evolution of species/tissue/mutation-specific mechanisms has also, perhaps, encountered some restrictions characteristic of cell types or cellular pathways in terms of the ability of evolution to prevent carcinogenesis.

In conclusion, based on the proposed revision to MMC theory, alongside some general tumor suppressive mechanisms, we should expect a multitude of species/group-specific cancer suppression strategies developed by different animal species. The evolution of species-specific mechanisms should foremost be driven by the evolution of species-specific longevity, body size and other life history traits that determine the relative contribution of health-related versus health-unrelated causes of mortality to the overall reproductive success as a function of chronological age. In this way, the overall age-dependent species-specific mortality curve impacts the contribution of somatic evolution and cancer to the overall reduction in reproductive success and thus determines the specific mode and strength of germline selection acting on the effects of particular somatic mutations on cellular physiology. It should be mentioned also that the curve of physiological aging, by imposing body frailty, is another major health-related contributor to age-dependent reduction in overall reproductive success. And like lethal cancers arising from somatic evolution, the species-specific physiological decline profile is forged by the same evolutionary force – the overall species-specific age-dependent mortality. It is therefore not surprising that increased cancer incidence in humans and captive animals mirrors the species-specific aging profile and occurs after the ages most individuals of the species survive in the wild (*Albuquerque et al., 2018*), as both processes are shaped largely by the same germline selection forces.

We therefore propose that the above-described theoretical paradigm explains how evolution at the population level shapes the impact of each specific driver mutation on particular tissues of particular species in order to explain the current body of knowledge on cancer incidence. We argue that differential aging-dependent somatic selection in cooperation with the group-specific evolution of particular genes, as well as cellular machinery in general, that depends on the evolution of life history traits are sufficient to generalize the process of carcinogenesis such that even phenomena such as early peaks in childhood cancer incidence of some cancers, Peto's paradox, and the universal life-span-dependent pattern of cancer incidence can be explained within one theoretical paradigm that places the theory of somatic evolution within the framework of general evolutionary theory.

## Materials and methods

### Software

All simulations were performed in the Matlab environment (MathWorks Inc., Natick, Massachusetts). We built a Monte Carlo simulation model that operates with a pool of simulated cells recapitulating the dynamics of stem cells (SC) in a competing self-renewing SC compartment.

### Model architecture

The model is based on the previously published general model of the human HSC dynamics (*Rozhok et al., 2016*; *Rozhok et al., 2014*). The simulated cells form one large competing population within a limited SC niche space, which has been argued as a suitable model for HSCs shown to migrate extensively and compete for bone marrow SC niches throughout the body (*Abkowitz et al., 1996*; *Abkowitz et al., 2000*; *Wright et al., 2001*). The simulation starts with a small pool of 300 cells which grows by the simulated age of 18–20 years up to 10,000 cells, following estimates by *Catlin et al. (2011)* and *Abkowitz et al. (2002)*. Estimates for the number of human HSCs range within $\sim\!10^4$ to $3 \times 10^5$, however our model investigated relative clonal dynamics of somatic mutant under different assumptions, therefore the exact adult size of the simulated pool is not critically important. The beginning pool of 300 cells has also been argued for early postnatal human HSC (*Abkowitz et al., 1996*; *Abkowitz et al., 2002*) (*Figure 2B*, upper chart). It is presently unknown whether this size is an accurate estimate. Nevertheless, we reasoned that in total, the human HSC pool does start from a small number of cells, even if the 300 estimate occurs pre-natally, leaving the exact number largely irrelevant for the purpose of the present study. The general principle, however, was held in the model that the HSC pool increases dramatically during body growth and maturation. The simulation was updated each simulated 'week' (the basic model update step) and lasted through 5200 updates, simulating thus a lifespan of 100 years.

### We simulated two principal age-dependent cell division profiles

The first one kept the average cell division rate stable throughout the entire simulation at $\sim\!1$ division in 20 weeks. This was done in order to test the assumptions made by Armitage and Doll (*Armitage and Doll, 1954*) at the time when the MMC model was created and data on stem cell behavior were not available. The second regimen in our simulation reflects modern data on HSC division rates showing a dramatic slowdown of HSC division activity by maturity (*Bowie et al., 2006*; *Sidorov et al., 2009*) (*Figure 2B*, lower chart), supported also by data demonstrating a similar slowdown in the accumulation of epigenetic change (*Horvath, 2013*). Following this paradigm, the average simulated cell division rates started from $\sim\!1$ division in 3 weeks and reached the adult rate of $\sim\!1$ division in 40 weeks (*Catlin et al., 2011*). We tested the effect of adult cell division rate in a wider range, from $\sim\!1$ division in 40 weeks to $\sim\!1$ in 70 weeks in order to explore other published estimates for HSC. The two regimens were explored in order investigate if such a departure from the initial MMC assumption of linear accumulation of damage can influence the modeled HSC clonal behavior. Cell divisions occurred stochastically by comparing each cell's time past the last division to the average division rate at any specific simulated age, following the general Gillespie algorithm (*Gillespie, 1977*), whereby the time past division of a specific cell was compared to the time generated from a normal distribution with the mean equal to the average cell division rate specific to the current simulated age and a standard deviation equal to mean/8, as previously argued (*Rozhok et al., 2014*).

Initial somatic fitness for all cells was set to 1. After each cell division, a driver mutation could occur based on a probability referred to here as the *phenotypic mutation rate*. The phenotypic mutation rate is higher than typically expected mutation rates for DNA base pair substitutions, as it integrates all possible changes (including epigenetic) in a cell that can endow the cell with a significant heritable alteration in somatic fitness. In the traditional MMC setup, such a driver mutation confers the recipient cell a certain constant fitness advantage over the normal cells. Each successive driver mutation increases fitness further, following the classic MMC paradigm (*Bozic et al., 2010*; *Tomasetti et al., 2015*). Such fitness alterations are age-independent. For simplicity, we assigned each driver mutation the same driving potential. An alternative model for the effect of somatic driver mutations was based on the principle of altering somatic selection, stating that the somatic fitness

value of somatic driver mutations depends on physiological aging (*DeGregori, 2013*; *Rozhok and DeGregori, 2015*). Within the latter paradigm, driver mutations are negatively selected early in life and positively selected during post-reproductive ages. This extension of MMC, designated as MMC-DS, provides that the somatic fitness effects of somatic driver mutations dynamically change over lifetime, as shown in *Figure 2C*, as a function of physiological aging. We tracked cell clones containing 1 through four driver mutations in order to investigate the relative timing and age-dependent clonal size curves of clones that require a certain minimum number of drivers. As such, the 1-through 4-mutation clones shown in the Results represent cell counts for clones that contain, respectively, 1 + through 4 + driver mutations. Mutation effects are defined by the equation (current fitness) = (initial fitness +mutation fitness effect)(number of mutations). Therefore, the combined effect of multiple phenotype altering mutations is synergistic rather than additive. The fitness of cells that did not acquire somatic driver mutations was left unaltered.

After division and mutation, all cells were subjected (at each 'weekly' simulated step) to a binomial trial in which the probability of each cell's survival in the simulated self-renewing pool depended on the current pool capacity, the number of competing cells after all divisions, and the cell's somatic fitness relative to other cells. In this way, our modeled somatic fitness parameter reflected what it is in natural stem cells - the ability of a cell to remain and proliferate in the self-renewing tissue compartment as opposed to leaving it by committing to differentiation, senescing or dying. A graphic representation of all the processes during one model update is shown in *Figure 2A*. As a result of the simulated cell division, mutation and competition, we observed the age-dependent changes in the representation of different mutant clones in the simulated pool. The exact algorithm of the simulation can be seen in Supplementary Materials section Model Matlab Code.

## Definitions

Hereby we will use the term *somatic evolution* as changes in the clonal composition of self-renewing tissue cell compartments of the body during an individual's lifespan. *Somatic selection* is defined as a process of differential maintenance of particular cell phenotypes within the self-renewing cell compartment of a tissue based on the particular *somatic fitness. Somatic drift* will be understood as the process of differential survival and proliferation of particular cell phenotypes driven by stochastic processes independent of cell somatic fitness in the self-renewing cell compartment of a tissue. *Cellular somatic fitness* is defined as the ability of a particular cell phenotype to survive, proliferate and self-renew in the self-renewing cell compartment of a tissue as a result of somatic selection. *Germline selection* will be understood as selection at the organismal level, reflecting the process of differential survival of particular individual phenotypes in a population over generations based on individual fitness. We propose to discriminate the term *germline selection* from the process of differential survival of particular germ cell phenotypes in the self-renewing germ cell compartment, as this process represents a particular case of somatic selection acting on germ cells.

## Statistical analysis

Statistical comparisons of the simulated clonal dynamics were performed using the Matlab Statistics toolbox. We asked two principal questions: (1) within the same clone requiring the same minimum number of driver mutations, does alteration of the tested parameter influence the timing and shape of the age-dependent clonal dynamics? and 2) within the same value of the tested parameter, do clones requiring different minimum numbers (1, 2, 3 or 4) of driver mutations show different time/age-dependence? Each simulated condition (parameter values) were repeated with 50 independent simulation runs (unless otherwise indicated). Therefore, the resulting age-dependent clonal dynamics for each condition were represented with a pool of 50 time series (see *Figure 2—figure supplement 1*, top row for an example). In order to elucidate as much statistical information about the relative behavior of clones as possible, we applied the following statistical procedure. At each timepoint (out of the 5200 total simulation timepoints), we compared different conditions each represented by a sample of 50 runs by the Kruskal-Wallis method, which is a non-parametric version of ANOVA. The obtained p-values were plotted along the X-axis (simulation timepoints), with the Y-axis representing p-values (see *Figure 2—figure supplement 1*, bottom row). This procedure allows visualizing the temporal dynamics of the differences in clonal behavior. The general magnitude of the difference in clonal behavior over time in this way can be visualized by the total sum of p-values (area under the

p-value curve). We calculated this area and divided it by the total area of the chart, the latter being $1 \times 5200$. The total area represents a hypothetical scenario whereby p-values are equal to one during an entire simulation, meaning that the compared behavior of clones was identical throughout the simulation. Respectively, if the area under the p-value curve equals zero, it would mean that such clonal behaviors are totally distinct throughout the simulation time. Realistically, however, p-values always are within that range and never reach such extremes. Therefore, the above-mentioned ratio shown in the top right corner of the chart in *Figure 2—figure supplement 1*, bottom row, and in *Figure 3—figure supplements 1–3* and *Figure 4—figure supplements 1–3*, reflects the overall relative magnitude of the difference in clonal behavior throughout the compared simulations. The smaller the ratio, the greater the overall difference in clonal behavior. Following this statistical procedure, thus, we can demonstrate both the significance of the difference at each timepoint (p-value curve) and the overall magnitude of the difference throughout the simulation time.

## Acknowledgements

The authors thank Brian Ross of the University of Colorado for valuable comments on the manuscript. These studies were supported by National Cancer Institute grant R01CA180175 and by the Courtenay C and Lucy Patten Davis Endowed Chair in Lung Cancer Research to JD.

## Additional information

### Funding

| Funder | Grant reference number | Author |
| --- | --- | --- |
| National Cancer Institute | R01CA180175 | James DeGregori |
| University of Colorado Boulder | University of Colorado Foundation, Courtenay C. and Lucy Patten Davis Endowed Chair in Lung Cancer Research | James DeGregori |

The funders had no role in study design, data collection and interpretation, or the decision to submit the work for publication.

### Author contributions

Andrii Rozhok, Conceptualization, Data curation, Software, Formal analysis, Investigation, Methodology, Writing—original draft, Writing—review and editing; James DeGregori, Conceptualization, Data curation, Funding acquisition, Project administration, Writing—review and editing

### Author ORCIDs

Andrii Rozhok http://orcid.org/0000-0002-9475-0472
James DeGregori http://orcid.org/0000-0002-1287-1976

### Decision letter and Author response

Decision letter https://doi.org/10.7554/eLife.39950.020
Author response https://doi.org/10.7554/eLife.39950.021

## Additional files

### Supplementary files

• Supplementary file 1. Model Matlab code.
DOI: https://doi.org/10.7554/eLife.39950.015

• Transparent reporting form
DOI: https://doi.org/10.7554/eLife.39950.016

## Data availability

Primary data related to this submission are publicly available at Dryad: https://dx.doi:10.5061/dryad.6ff5k07

The following dataset was generated:

| Author(s) | Year | Dataset title | Dataset URL | Database and Identifier |
|---|---|---|---|---|
| Rozhok A | 2019 | Data from: A generalized theory of somatic evolution | http://dx.doi.org/10.5061/dryad.6ff5k07 | Dryad Digital Repository, 10.5061/dryad.6ff5k07 |

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
