## [Decision Letter]

[**Editorial note:** This article has been through an editorial process in which the authors decide how to respond to the issues raised during peer review. The Reviewing Editor's assessment is that all the issues have been addressed.]

Thank you for submitting your article "A generalized theory of somatic evolution" for consideration by *eLife*. Your article has been reviewed by two peer reviewers, and the evaluation has been overseen by Eduardo Franco as Reviewing Editor and Aviv Regev as the Senior Editor. The reviewers have opted to remain anonymous.

The Reviewing Editor has highlighted the concerns that require revision and/or responses, and we have included the separate reviews below for your consideration. In the interest of providing you with constructive feedback we have edited the reviewers' critiques while keeping the substantive points in need of revision. If you have any questions, please do not hesitate to contact us.

Your modelling work advances our understanding of multi-stage carcinogenesis by accommodating the effects of ageing and other potential drivers of biological variability. However, reviewers have raised significant major concerns. In brief, they challenged many of your assumptions regarding biological mechanisms and were disappointed with the lack of details on the methodology and with the sanguine interpretation of model outputs without firm support on tenable assumptions.

We strongly encourage you to take the critiques below very seriously. The value of your paper to *eLife* readers will substantially increase if you are able to accommodate the many concerns that are transcribed below.

When resubmitting, please revise the title of your paper to reflect its focus on carcinogenesis or prediction of age-specific cancer incidence.

Separate reviews (please respond to each point and revise accordingly):

*Reviewer #1:*

I worry that many of the statements or presentations in this submission are imprecise or technically sloppy. These statements must be amended and backed by logic, a detailed explanation of the methods, and/or references to prior work.

In subsection “Quick guide to model”, this is written: "We simulate the assumed fixed fitness effects following the assumptions made in many modeling studies and by modern MMC theorists (Bozic et al., 2010; Tomasetti et al., 2015; Vogelstein et al., 2013)." Even if that point is clarified elsewhere in the paper, that statement is much too vague there to be acceptable.

In Figure 2A, the following is written: "Cells divide stochastically based on the last time a specific cell divided and the age-dependent cell division rate" and "Cell's relative chances of staying in the self-renewing pool over time depend on the niche space available, the number of competing cells and the cell's relative somatic fitness." Those statements in that figure are mathematically imprecise and unacceptable. They don't allow anything to be independently reproduced. Please revise to permit independent verification.

The Matlab code dump in Supplementary file 1 lacks details. The authors must provide enough explanatory text to explain their programming code. As it stands the presentation of the modeling procedures is rather cryptic. The discussions in the main text must be based on clear and detailed annotations to the code. The authors must create a dedicated section in Supplementary file 1 that is carefully and pedagogically written to explain the simulation steps.

Are the vertical axis scales in Figure 3 linear or logarithmic? Are the vertical axis scales in Figure 4 linear or logarithmic? In Figure 5, what is the numbering on the horizontal axes?

*Reviewer #2:*

This is a well-written and reasoned paper that develops and tests a mathematical model for understanding the observed rates of cancer incidence across diverse tissues/organs & species. Whilst I am not a mathematical modeller and am only superficially familiar with Matlab, I find the approaches used sensible and the assumptions underlying them valid overall. In my view the most important contribution of this manuscript is to highlight the non-cell-autonomous effect of ageing on somatic evolution and demonstrate how taking this into account can help better explain the observed cancer incidence. However, some of the assumptions made and inferences drawn will be open to criticism and also some known variables are not tested in the proposed models. Below I make some suggestions about how this could be addressed and also some other comments.

Major comments:

The assumption that the fitness advantage imparted by a mutation is constant is likely to be untrue in some instances. As HSCs are used as the model here, clonal haematopoiesis (CH) is a pertinent example. In CH, clonal expansions do not inexorably lead to clonal growth over time and in many instances clones that expand initially, stop expanding or even shrink later (e.g. Young et al., Nat Comms, 2016 and Abelson et al., Nature, 2018). It would be worth modelling the impact of this: i.e. will clonal behaviour differ if the fitness advantage fluctuates between 1.0 and 1.1 at different times, whilst averaging 1.05 (vs a constant advantage of 1.05).

Similarly, HSC or other stem cell divisions may be driven by intermittent life events (such as intercurrent illnesses /infections). Would such a fluctuating behaviour alter predictions or influence effects of drift in the author's models?

The concept of molecular synergy between mutations is not explored and this is also true of the concept of cellular transformation (often a consequence of powerful synergy). These are both well-established concepts operating in the evolution to CH to acute leukaemia. The most significant way in which these phenomena could influence the proposed models, is by achieving extremes of fitness advantage that go beyond the ranges explored here. These phenomena have been invoked to explain observations such as the peak in ALL incidence in early childhood in association with activated mutational processes (e.g. Swaminathan et al., Nat Immunol, 2015). These phenomena should at least be discussed if not modelled.

Minor comments:

The term "life history traits" should be explained/discussed early in the manuscript and in the context of life history theory. e.g. list some of these traits and mention that they are seen as determinants (through natural selection) of diverse organismal characteristics of a species.

The interaction between ageing and clonal expansion has been previously reported for clonal haematopoiesis in association with particular mutations (e.g. McKerrell et al., 2015) and a brief mention of this in Introduction would be helpful to readers.

The authors discuss the impact of ageing on clonal selection solely form the point of you of the ageing environment or niche. The potential impact of cell-intrinsic ageing of tissue stem cells should also be mentioned even if this does not affect modelling.

Estimates of the number of HSCs are controversial. The authors explain in Materials and methods that the specific number used here would not affect their model. In the relevant Results section, it would be worth referring readers to methods for this explanation. Similarly, the fact that the authors do take into account mutations other than nucleotide substitutions and also heritable epigenetic change should also be mentioned or alluded to before Materials and methods.

In the Introduction and Quick guide to model sections the authors discuss that "~50% of mutations accumulate before maturity". This is reportedly not the case in HSCs, which appear to harbour very few somatic variants in the first two decades of life (Welch et al., 2012). There is also evidence that in some individuals, mutational processes driven by DNA editing enzymes may augment the number of mutations. As HSCs are used as the model here, the authors should discuss this and mention whether/how it affects their calculations. Are they suggesting that purifying selection removes some/most of the more mutated HSCs If so, this would be controversial and thus requires more explanation or rethinking.

In the Introduction the phrase "… approximately the same age of incidence increase" is not clear and should be made clearer – e.g. "… approximately the same fractional increases in incidence with age".

In subsection “Quick guide to model” paragraph two, the statement "… always increase cellular somatic fitness" appears redundant when a driver mutation's driving potential is constant.

Figure 2A: legend should make it clear that the model refers to fate choices for a single cell.

Figure 2B: readers will notice that at age 0 the number of cells is very small (~1 cell). This is probably because age 0 refers to the fertilised zygote. This should be explained to avoid confusion.

Figure 3B: light and dark blue lines appear to have been switched in legend.

---

## [Author Response]

Your modelling work advances our understanding of multi-stage carcinogenesis by accommodating the effects of ageing and other potential drivers of biological variability. However, reviewers have raised significant major concerns. In brief, they challenged many of your assumptions regarding biological mechanisms and were disappointed with the lack of details on the methodology and with the sanguine interpretation of model outputs without firm support on tenable assumptions.We strongly encourage you to take the critiques below very seriously. The value of your paper to eLife readers will substantially increase if you are able to accommodate the many concerns that are transcribed below.When resubmitting, please revise the title of your paper to reflect its focus on carcinogenesis or prediction of age-specific cancer incidence.

We modified the title as requested to “A generalized theory of age-dependent carcinogenesis”.

Separate reviews (please respond to each point and revise accordingly):

Reviewer #1:

I worry that many of the statements or presentations in this submission are imprecise or technically sloppy. These statements must be amended and backed by logic, a detailed explanation of the methods, and/or references to prior work.In subsection “Quick guide to model”, this is written: "We simulate the assumed fixed fitness effects following the assumptions made in many modeling studies and by modern MMC theorists (Bozic et al., 2010; Tomasetti et al., 2015; Vogelstein et al., 2013)." Even if that point is clarified elsewhere in the paper, that statement is much too vague there to be acceptable.

We changed this and other statements to make them clearer.

In Figure 2A, the following is written: "Cells divide stochastically based on the last time a specific cell divided and the age-dependent cell division rate" and "Cell's relative chances of staying in the self-renewing pool over time depend on the niche space available, the number of competing cells and the cell's relative somatic fitness." Those statements in that figure are mathematically imprecise and unacceptable. They don't allow anything to be independently reproduced. Please revise to permit independent verification.

The first statement was changed. The second statement was not meant to be mathematically precise, as it is just a general description of the principle. Neither of the statements was meant to provide a guide for how to reproduce the results – the specific algorithms are too cumbersome to be put into a figure. Therefore, those are provided in the supplements in the Matlab code of the model. We now use the steps in Figure 2A as guideposts throughout the Matlab code provided in the supplement, so that a reader can better appreciate the math underlying each step.

The Matlab code dump in the Supplementary file 1 lacks details. The authors must provide enough explanatory text to explain their programming code. As it stands the presentation of the modeling procedures is rather cryptic. The discussions in the main text must be based on clear and detailed annotations to the code. The authors must create a dedicated section in the Supplementary file 1 that is carefully and pedagogically written to explain the simulation steps.

We have added more annotation to guide the reader through the code as suggested. As mentioned above, we also now use Figure 2 as the guide for different steps in the code. We appreciate this critique, as such annotation will aid those wishing to reproduce or extend our modeling.

Are the vertical axis scales in Figure 3 linear or logarithmic? Are the vertical axis scales in Figure 4 linear or logarithmic? In Figure 5, what is the numbering on the horizontal axes?

The numbering and scales are now explained in the legend and are identical for all charts. We added clarification that the scales are linear.

Reviewer #2:

This is a well-written and reasoned paper that develops and tests a mathematical model for understanding the observed rates of cancer incidence across diverse tissues/organs & species. Whilst I am not a mathematical modeller and am only superficially familiar with Matlab, I find the approaches used sensible and the assumptions underlying them valid overall. In my view the most important contribution of this manuscript is to highlight the non-cell-autonomous effect of ageing on somatic evolution and demonstrate how taking this into account can help better explain the observed cancer incidence. However, some of the assumptions made and inferences drawn will be open to criticism and also some known variables are not tested in the proposed models. Below I make some suggestions about how this could be addressed and also some other comments.Major comments:The assumption that the fitness advantage imparted by a mutation is constant is likely to be untrue in some instances. As HSCs are used as the model here, clonal haematopoiesis (CH) is a pertinent example. In CH, clonal expansions do not inexorably lead to clonal growth over time and in many instances clones that expand initially, stop expanding or even shrink later (e.g. Young et al., Nat Comms, 2016 and Abelson et al., Nature, 2018). It would be worth modelling the impact of this: i.e. will clonal behaviour differ if the fitness advantage fluctuates between 1.0 and 1.1 at different times, whilst averaging 1.05 (vs a constant advantage of 1.05).

A good point – the effect of mutation fitness effects, in fact, more often varies, rather than stay constant. For this reason, we apply the MMC-DS model with varying effects and test it against currently widely accepted paradigm that driver mutations have defined constant effects. Variations of the MMC-DS effects on fitness that we model include the strength of altered selection (both positive and negative), whether negative selection early in life mirrors positive selection late in life, and the age of onset of the switch in differential selection.

A multitude of additional scenarios are indeed possible. Publications by Christina Curtis’ group also show that later mutations in many cases behave as neutral, and late clonal dynamics in carcinogenesis follow neutral drift dynamics. However, in this study we intentionally avoided modeling specific (and hypothetical) scenarios as the purpose of the study was to compare two basically different paradigms – the classic MMC and its extension MMC-DS which assumes dominance of purifying selection early in life and positive selection for mutants late in life. The key question is whether MMC is sufficient generally to explain cross species distribution of cancer incidence, and if MMC-DS is capable of improving the result by adding age-dependent selection. We find that MMC-DS does improve over MMC, but still does not answer all questions. For this reason, we propose an additional theoretical framework of life history dependent species-specific evolution of genes and thus that the combined effects of the MMC-DS paradigm and life history dependent evolution of genes can universally explain cancer incidence across the animal kingdom. We did not intend to go beyond this focus. Still, we now describe in Quick guide to model how the fitness effects of mutations could be even more complicated than as modeled in the MMC-DS conditions, such as through reversal of a fitness advantage.

Similarly, HSC or other stem cell divisions may be driven by intermittent life events (such as intercurrent illnesses /infections). Would such a fluctuating behaviour alter predictions or influence effects of drift in the author's models?

These scenarios are also interesting in the context of further exploration of the model, which we did not intend to do for the present study. There are a multitude of different fluctuations, timing, amplitude and periodicity (or randomness) of these fluctuations, and thus we would need to make somewhat arbitrary choices. These changes however would not impact the overall question that we have focused on – does MMC have to incorporate changing age-dependent selection through life and whether such altered selection is sufficient to make MMC a generalistic cross-cancer and cross-species model.

The concept of molecular synergy between mutations is not explored and this is also true of the concept of cellular transformation (often a consequence of powerful synergy). These are both well-established concepts operating in the evolution to CH to acute leukaemia. The most significant way in which these phenomena could influence the proposed models, is by achieving extremes of fitness advantage that go beyond the ranges explored here. These phenomena have been invoked to explain observations such as the peak in ALL incidence in early childhood in association with activated mutational processes (e.g. Swaminathan et al., Nat Immunol, 2015). These phenomena should at least be discussed if not modelled.

The reviewer is correct about synergism among mutations, specifically known for leukemia. However, the conclusion made by the reviewer was probably confused by our poor annotation of the original code. We added more extended annotation. Our model does operate primarily with the synergistic type of effect in the sense that fitness increase is not additive. The equation of cell fitness is fitness = (1 + fitness effect of a single mutation)^number of mutations. E.g. if the effect of a single mutation is 20% and a cell has accumulated 3 mutations, the cell’s fitness will be (1+0.2)^3 = 1.2x1.2x1.2 = 1.728 (not 1.6). Meaning that each next mutation amplifies the effect of preceding mutations. Moreover, this type of interaction between fitness-altering mutation is known more widely in evolution, beyond carcinogenesis. We hope that the way fitness is determined in the model is now more apparent in the code. We have also added explanation of this mutational synergism to the Materials and methods section.

Minor comments:The term "life history traits" should be explained/discussed early in the manuscript and in the context of life history theory. e.g. list some of these traits and mention that they are seen as determinants (through natural selection) of diverse organismal characteristics of a species.

We added examples of life history traits where the term is first used in the Introduction. The effect of life history traits on somatic evolution is further discussed in the Discussion where we propose a theory of how life history links with somatic evolution.

The interaction between ageing and clonal expansion has been previously reported for clonal haematopoiesis in association with particular mutations (e.g. McKerrell et al., 2015) and a brief mention of this in Introduction would be helpful to readers.

This is a good point, since we model clonal evolution in general. We added several related citations, including the one recommended by the reviewer.

The authors discuss the impact of ageing on clonal selection solely form the point of you of the ageing environment or niche. The potential impact of cell-intrinsic ageing of tissue stem cells should also be mentioned even if this does not affect modelling.

A good point as well. We added a discussion of this at the end of the Quick guide to model section where the MMC-DS paradigm is justified.

Estimates of the number of HSCs are controversial. The authors explain in Materials and methods that the specific number used here would not affect their model. In the relevant Results section, it would be worth referring readers to methods for this explanation. Similarly, the fact that the authors do take into account mutations other than nucleotide substitutions and also heritable epigenetic change should also be mentioned or alluded to before Materials and methods.

Statement added in the Results section, with references to papers showing a range of HSC numbers from ~10,000 to several hundred thousands in humans. And explanation on mutations is added to the Quick guide to model section.

In the Introduction and Quick guide to model sections the authors discuss that "~50% of mutations accumulate before maturity". This is reportedly not the case in HSCs, which appear to harbour very few somatic variants in the first two decades of life (Welch et al., 2012). There is also evidence that in some individuals, mutational processes driven by DNA editing enzymes may augment the number of mutations. As HSCs are used as the model here, the authors should discuss this and mention whether/how it affects their calculations. Are they suggesting that purifying selection removes some/most of the more mutated HSCs If so, this would be controversial and thus requires more explanation or rethinking.

The data generated by Welch et al. markedly stand out from multiple other publications that demonstrate that roughly half of mutations accumulate early in life. We would rather not go into a critique of the Welch et al. methods in this manuscript, but we do note in particular that the mutation accumulation patterns derived are based on very small numbers of people, HSPCs examined, and mutations identified. We are not suggesting that stabilizing selection removes most mutations, as most somatic mutations will be neutral. Notably, a very large dataset obtained by Steve Horvath (PMID: 24138928) for human tissues, in particular, shows that in the hematopoietic system early accumulation of epigenetic changes is the case. Other cited examples from mice (from the Vijg group) also show early life accumulation of a large fraction of mutations. The reason for such accumulation is that early in life HSCs in particular have very high division rates while the body is growing (e.g. Bowie et al., PMID: 17016561). In the model, we did not make any assumptions on mutation accumulations, the latter is the result of the age-dependent cell division rate profile shown in Figure 2B. Regardless, we now reference Welch et al. and indicate that the early life pattern of mutation accumulation is not universally observed. Regarding variation in mutation rate among individuals, we vary the mutation rate (Figure 3C), and demonstrate that the MMC-DS model is more robust to this variation (relative to the MMC model). We also simulate both a dynamic cell division profile (resulting in more mutations occurring early in life) and a stable cell division profile (which would lead to more linear mutation accumulation with age); notably, these changes in division profile minimally impacted clonal dynamics in the model.

In the Introduction the phrase "… approximately the same age of incidence increase" is not clear and should be made clearer – e.g. "… approximately the same fractional increases in incidence with age".

Corrected as suggested by the reviewer.

In subsection “Quick guide to model” paragraph two, the statement "… always increase cellular somatic fitness" appears redundant when a driver mutation's driving potential is constant.

We emphasized this point to make sure the reader understands that the effect is increasing fitness, as opposed to a possible constant effect that decreases fitness.

Figure 2A: legend should make it clear that the model refers to fate choices for a single cell.

Clarification added as suggested.

Figure 2B: readers will notice that at age 0 the number of cells is very small (~1 cell). This is probably because age 0 refers to the fertilised zygote. This should be explained to avoid confusion.

The simulation started from 300 cells in settings with the adult pool size of 10,000 cells, and with proportionally larger initial pool size in simulations with larger adult pool sizes. We added clarification to the legend as suggested.

Figure 3B: light and dark blue lines appear to have been switched in legend.

Thanks for catching this. Corrected as suggested.